# Probiotics as an Alternative to Antibiotics: Genomic and Physiological Characterization of Aerobic Spore Formers from the Human Intestine

**DOI:** 10.3390/microorganisms11081978

**Published:** 2023-07-31

**Authors:** Maria Vittoria, Anella Saggese, Rachele Isticato, Loredana Baccigalupi, Ezio Ricca

**Affiliations:** 1Department of Biology, Federico II University of Naples, 80125 Naples, Italy; maria.vittoria@unina.it (M.V.); anella.saggese@unina.it (A.S.); isticato@unina.it (R.I.); 2National Biodiversity Future Center (NBFC), 90133 Palermo, Italy; lorbacci@unina.it; 3Department of Molecular Medicine and Medical Biotechnology, Federico II University of Naples, 80131 Naples, Italy

**Keywords:** probiotics, antimicrobials, beneficial bacteria, *Bacillus velezensis*, *Bacillus subtilis*, *Priestia megaterium*, antibiotic resistance, antioxidants, CSF

## Abstract

A total of thirty-two aerobic spore former strains were isolated from intestinal samples of healthy children and analyzed for their hemolytic and antibiotic-resistant activities. Four strains selected as non-hemolytic and sensitive to all antibiotics recommended as relevant by regulatory agencies were short-listed and evaluated for their in silico and in vitro probiotic potentials. The four selected strains were assigned to the *Bacillus velezensis* (MV4 and MV11), *B. subtilis* (MV24), and *Priestia megaterium* (formerly *Bacillus megaterium*) (MV30) species. A genomic analysis indicated that MV4, MV11, and MV24 contained a homolog of the gene coding for the fibrinolytic enzyme nattokinase while only MV30 encoded a glutamic acid decarboxylase essential to synthesize the neurotransmitter GABA. All four strains contained gene clusters potentially coding for new antimicrobials, showed strong antioxidant activity, formed biofilm, and produced/secreted quorum-sensing peptides able to induce a cytoprotective stress response in a model of human intestinal (HT-29) cells. Altogether, genomic and physiological data indicate that the analyzed strains do not pose safety concerns and have in vitro probiotic potentials allowing us to propose their use as an alternative to antibiotics.

## 1. Introduction

Antibiotic resistance is a global health threat, predicted to cause, by 2050, a million annual deaths due to an increased incidence of infectious diseases that will, in turn, affect many medical and surgical procedures [1]. So far, the identification and design of new antibiotics, and the development of novel vaccines and probiotics have been proposed as alternatives to current antibiotics [2]. Probiotics, live bacteria that exert a beneficial health effect, are recognized as potential candidates to substitute antibiotics for the capacity of some bacteria to protect the animal body against bacterial and viral infections. Inhibition of bacterial adhesion, enhanced mucosal barrier function, modulation of the immune systems, and secretion of antimicrobial metabolites are the main mechanisms proposed to explain the anti-infectious activity of probiotics [3].

Several species of the *Bifidobacterium*, *Lactobacillus*, and *Bacillus* genera have long been used as commercial probiotics [4], while other bacteria, such as *Faecalibacterium prausnitzii* and *Akkermansia muciniphila* have been recently proposed as next-generation probiotics [5,6]. In this context, the case of *Bacillus*-based probiotics is peculiar since all commercial preparations do not contain vegetative cells but metabolically quiescent spores [7]. For these organisms, spore formation is a survival strategy since spores are extremely stable cells, resistant to harsh conditions such as high temperatures, low pH, absence of water and nutrients, presence of lytic enzymes, toxic chemicals, and UV irradiations [7]. In response to the renewed presence of water, nutrients, and of favorable environmental conditions, spores germinate, originating vegetative cells able to grow and eventually sporulate again [8]. Also, due to their resistance properties, *Bacillus* spores are ubiquitous in nature and can be isolated from common and extreme environments [7,8]. *Bacillus* spores are also abundantly present in the gut of animals, where they enter as contaminants of food and water and safely transit the gastric barrier. In the intestine, some of the ingested spores germinate [9,10,11,12], and the germination-derived cells grow and temporarily colonize that niche, before re-sporulating in the terminal part of the intestine and leaving the body as spores [13,14].

Ingested spores of various *Bacillus* species have health-beneficial effects [9,11] and provide protection against bacterial [15] and viral infections [16]. Such probiotic effects have been ascribed to various mechanisms, including the promotion of the development of the Gut-Associated Lymphoid Tissue (GALT) [17], the induction of the production of cytokines in mesenteric lymph nodes (MLN) (IL-1, IL-5, IL-6, IFN-γ, and TNF-α) and spleen (IFN-γ and TNF-α) [18], the upregulation of the expression of the Toll-like receptors TLR-2 and TLR-4 [19], and the protection of human intestinal cells from oxidative stress [20] by inducing the nuclear translocation of the transcriptional factor Nrf-2 that, in turn, activates stress-response genes [21]. More recently, *B. subtilis* cells have been shown to have regulatory effects on epithelial differentiation by inhibiting the Notch pathway and, as a consequence, shifting differentiation toward a secretory cell fate [22]. Such a Toll-like receptor 2-dependent effect causes an increased number of Paneth and goblet cells in the intestine that, in turn, results in the production of antimicrobial peptides [22]. An additional mechanism by which spores exert their beneficial effects is the modulation of the microbial composition of the gut, exerted by favoring the prevalence of other potentially beneficial bacteria such as *Faecalibacterium prausnitzii* [23], *Akkermansia muciniphila*, and *Bifidobacterium* spp. [24].

In the intestine, germination-derived cells produce a variety of metabolites that mediate the interaction with the host’s intestinal and immune cells. Examples of such beneficial molecules produced by several *Bacillus* strains are enzymes, such as the NattoKinase (NK), a serine protease characterized by a potent thrombolytic activity and proposed for the treatment of multiple cardiovascular diseases [25], and the Glutamic Acid Decarboxylase (GAD), able to catalyze the synthesis of γ-aminobutyric acid (GABA), the most common inhibitory neurotransmitter in the central nervous system that also regulates cardiovascular functions such as blood pressure and heart rate [26]. In addition, peptides and lipopeptides with antimicrobial activity [27,28] and quorum-sensing peptides [29] are also produced by several *Bacillus* strains. An example is the Competence and Sporulation Factor (CSF), a quorum-sensing pentapeptide secreted by *B. subtilis* cells that is also recognized by human epithelial cells in which it induces the synthesis of the heat-shock (HS) proteins [29]. In a murine model, CSF prevents oxidant-induced intestinal epithelial injuries and loss of barrier function, exerting a cytoprotective activity [30]. CSF is only produced by members of the *B. subtilis* species but other species produce different peptides (CSF-like) similarly able to induce the synthesis of the heat-shock (HS) proteins in human cells in vitro [31]. More recently, CSF has been recognized as a key factor for the anti-neurodegenerative activity of *B. subtilis* in the model organism *Caenorhabditis elegans* [32]. *B. subtilis* strains able to produce biofilm, CSF, and nitric oxide (NO) were shown to extend the longevity of *C. elegans* by downregulating the insulin-like signaling system, thus opening the possibility that ingested spores of *B. subtilis* could contribute to delay the development of age-related diseases [33,34]. 

In this work, several *Bacillus* strains were isolated from intestinal samples of healthy children not under probiotic or antibiotic treatment. Four of these strains which were not hemolytic and not resistant to antibiotics, were assigned to the *B. velezensis* (MV4 and MV11), *B. subtilis* (MV24), and *Priestia megaterium* (formerly *B. megaterium*) (MV30) species and characterized at the genomic and physiological level.

## 2. Materials and Methods

### 2.1. Bacterial Isolation and Characterization

Fecal samples of healthy children (from ten months to ten years old) that had not been under antibiotic or probiotic treatment in the previous three months were collected and about 1 g of feces homogenized in sterile Phosphate-buffered saline (PBS) and heat-treated at 80 °C for 20 min. The samples were then serially diluted and plated on Difco Sporulation Medium (DSM, for 1 L: 8 g/L Nutrient Broth, 1 g/L KCl, 1 mM MgSO_4_, 1 mM Ca(NO_3_)_2_, 10 μM MnCl_2_, 1 μM FeSO_4_, Sigma-Aldrich, Darmstadt, Germany) and incubated overnight at 37 °C. The colonies were selected and isolated as previously described [35]. All isolates were tested for hemolytic activity by spotting 10 µL of growing cells on Columbia Agar plates supplemented with 5% defibrinated horse blood (ThermoScientific, Waltham, MA, USA) and, after 24 h of incubation at 37 °C, the clear halo around the colonies indicated a positive result. The non-hemolytic strains were then tested for their antibiotic susceptibility by plating 0.5 McFarland of bacteria on LB agar plates (for 1 L: 10 g/L Bacto-Tryptone, 5/L g Bacto yeast extract, 10 g/L NaCl, pH 7.0). Antibiotic-impregnated MIC test strips (Liofilchem^®^, Roseto, Italy) were then placed on the plates and incubated overnight at 37 °C. The minimum inhibitory concentration (MIC) was measured at the intersection of the inhibition halo and the antibiotic strip. The strains were indicated as resistant (R) if the antibiotic concentration exceeded the threshold level set by the EFSA for *Bacillus* strains [36].

### 2.2. Whole-Genome Sequencing and Bioinformatic Analysis

Bacterial genomic DNA was extracted from exponentially growing cells as previously reported [37]. Genome sequencing of MV4, MV11, MV24, and MV30 was performed with Illumina MiSeq Sequencing System by GenProbio (Parma, Italy). Genome assemblies were performed with SPAdes v3.14.0 by means of a MEGAnnotator pipeline [38]. The genomes of MV4, MV11, MV24, and MV30 have been deposited in GenBank as BioProject numbers PRJNA977871, PRJNA977867, PRJNA977868, PRJNA977866, and accession numbers CP127102, JASNWD000000000, JASPFH000000000, JASPFG000000000, respectively. For phylogenetic analyses, all genomic sequences were imported into the KBase system to create a genome set that was subsequently employed to build the tree, using Insert Genome Into SpeciesTree-v2.2.0. The phylogenomic trees were then visualized on the Interactive Tree of Life (iTOL, platform v6). Average Nucleotide Identity (ANI) values between the sequenced genomes and the closest bacteria were obtained using the Ezbiocloud tool (https://www.ezbiocloud.net/tools/ani (accessed on 3 October 2022)).

For the detection of toxin-associated genes, the genome of the strains was analyzed using the VF analyzer pipeline [39] as described above [35]. 

The presence of homologs of nattokinase and glutamate decarboxylase was investigated in aminoacid sequences of MV strains using protein BLAST alignment (https://blast.ncbi.nlm.nih.gov/Blast.cgi?PAGE=Proteins (accessed on 3 October 2022)).

Biosynthetic gene clusters for the secondary metabolites and predicted core structure peptides were obtained using the web-based genome mining tool antiSMASH (http://antismash.secondarymetabolites.org (accessed on 7 November 2022)). 

Functional annotations of MV genomes were obtained using EggNOG-mapper v2 (http://eggnog-mapper.embl.de, (accessed on 7 November 2022)). Similarly, the genomes were also scanned using the dbCAN2 server (https://bcb.unl.edu/dbCAN2/blast.php (accessed on 7 November 2022)) that integrated three tools/databases for automated CAZyme annotation [40].

### 2.3. Antimicrobial and Biofilm Production Assays

Antimicrobial activity was assessed as previously described [40]. Briefly, 10 µL of supernatant was spotted onto LB agar plates and the plates were air dried. Then, 100 µL of an exponential culture of each of the target strains was mixed with 10 mL of 0.7% LB agar and plated. Fresh medium was used as a negative control. The plates were incubated overnight at 37 °C and the antimicrobial activity was determined as the diameter of the inhibition halo. 

Biofilm formation was tested as previously reported [41]. Briefly, growing cells were inoculated into 24-well culture plates with three different media: LB, DSM [42], and S7 minimal medium (for 1 L: 50 mM morpholine-propane-sulfonic acid (MOPS) (adjusted to pH 7.0 with KOH), 10 mM (NH_4_)_2_SO_4_, 5 mM potassium phosphate (pH 7.0), 2 mM MgCl_2_, 0.9 mM CaCl_2_, 50 μM MnCl_2_, 5 μM FeCl_3_, 10 μM ZnCl_2_, 2 μM thiamine hydrochloride, 20 mM sodium glutamate, 1% glucose, 0.1 mg/mL phenylalanine, and 0.1 mg/mL tryptophan) [43]. The plates were incubated for 48 h at 37 °C, and then biofilm production was assessed by crystal violet assay as previously described in [41]. The experiment was performed in triplicate.

### 2.4. Antioxidant Activity Tests

#### 2.4.1. Hydrogen Peroxide Scavenging Assay

The hydrogen peroxide stability was measured by following absorbance at 240 nm of 1 mL of fresh hydrogen peroxide solution (50 mM Potassium Phosphate Buffer, pH 7.0; 0.036% (*w*/*w*) H_2_O_2_). Quantitative determination of H_2_O_2_ scavenging activity of cells of MV strains was measured by the loss of absorbance at 240 nm as previously described [20]. Briefly, 10^7^, 10^8^, and 10^9^ cells/mL of selected strains were incubated at room temperature in 1 mL of 0.036% of hydrogen peroxide solution. After 30 min, the hydrogen peroxide concentration in the supernatant was determined by measuring the absorbance at 240 nm. The percentage of peroxide removed was calculated as reported below:H_2_O_2 removed_ (%) = (1 − A_sample_/A_control_) × 100

#### 2.4.2. DPPH Assay

The α,α-diphenyl-β-picrylhydrazyl (DPPH) free radical scavenging method was used to evaluate the potential antioxidant activity of MV strains [20]. Different concentrations of the cells of each strain (10^7^, 10^8^, and 10^9^ cells/mL) were incubated in a final volume of 1 mL of 50 mM of citrate phosphate buffer (pH 7) containing 0.1 mM of freshly prepared DPPH (giving absorbance ≤ 1.0). The reaction was allowed to proceed for 30 min in the dark at room temperature. The DPPH free radical scavenging activity was then monitored by determining the absorbance at 515 nm and calculated according to the following equation:DPPH _radical scavenging activity_ (%) = (1 − A_sample_/A_control_) × 100
where A_sample_ is the absorbance of the reacted mixture of DPPH with the extract sample, and A_control_ is the absorbance of the DPPH solution.

### 2.5. Growth and Treatment of HT-29 Cells with Bacterial Conditioned Medium

HT-29 cells (ATCC HTB-38), derived from colon adenocarcinoma, were grown in RPMI 1640 medium (EuroClone S.p.A., Pero, Italy) supplemented with 10% fetal bovine serum and 1% penicillin-streptomycin. HT-29 cells were cultured at 37 °C in a humidified atmosphere of 5% CO_2_ [40]. At 80% confluence, cells were harvested and diluted in 6-well plates to a total concentration of 5.5 × 10^6^ cells per well. Bacterial conditioned medium (CM), required for the treatment of the HT-29 cells, was obtained by growing bacterial cells in minimal S7 medium, aerobically, for 16 h at 37 °C [44]. The culture was then centrifuged (7000 rpm for 10 min) and the supernatant (CM) was filtered through a 0.22 µm low protein binding filter (Millex, Millipore, Bedford, MA, USA). The obtained CM was used for HT-29 cells treatment at a concentration of 20% *v*/*v* in a complete growth medium as previously reported [44]. After 24 h incubation of HT-29 cells with 20% CM, the cells were harvested, lysed, and cell extracts were prepared for Western blot analysis as described below. A heat-shock treatment (45 °C for 30 min) was performed prior to cell lysis for heat-shock proteins induction in HT-29 cells (not treated with CM), used as a positive control. The experiment was repeated in triplicate.

### 2.6. Protein Extraction, SDS-PAGE, and Western Blot Analysis

Cells were scraped and harvested in lysis buffer (50 mM Tris-HCl pH 7.5, 5 mM EDTA, 150 mM NaCl, 1% NP-40, 1 mM phenylmethylsulfonyl fluoride, 0.5% sodium deoxycholate and protease inhibitors) and the cell lysates were incubated on ice for 40 min followed by centrifugation at 13,000 rpm for 30 min to remove cellular debris. Protein concentration was determined using the Bio-Rad protein assay (Bio-Rad, Segrate, Milano, Italy). Subsequently, 2X Laemmli buffer (Sigma Aldrich, Milan, Italy) was added and proteins were boiled at 100 °C for 5 min and separated on a 12% SDS-PAGE gel. Proteins were transferred (2 h at 100 V, 4 °C) to polyvinylidene fluoride membranes (Millipore, Burlington, MA, USA). The membranes were blocked in a PBS-Tween20 (0.2%) buffer, with 5% (*w*/*v*) Bovine Serum Albumin (BSA, PanReac AppliChem, Darmstadt, Germany) and incubated with the primary antibodies anti-HSP27 and anti-β-Actin (Elabscience^®^, Wuhan China), diluted 1:500 and 1:1000, respectively, in a PBS-Tween20-2.5% BSA solution overnight at 4 °C. After several washes with PBS-Tween20 buffer, the membranes were incubated with 1:7000 anti-rabbit secondary antibody (SantaCruz, Santa Cruz, CA, USA) for 2 h and then visualized by enhanced chemiluminescence (ECL; GE-Healthcare, Little Chalfont, UK) using horseradish peroxidase-conjugated secondary antibody (Santa-Cruz Biotechnology) and analyzed using Quantity One^®^ software from the ChemiDoc™ XRS system (version 4.62, Bio-Rad, Hercules, CA, USA). Band intensity was evaluated using ImageQuant analysis. Values were normalized to the β-Actin and band intensity was measured as follows: [Reference sample β-Actin/Treated sample β-Actin] × Band quantity (https://www.bio-rad.com/it-it/applications-technologies/image-analysis-quantitation-for-western-blotting?ID=PQEERM9V5F6X accessed on 6 February 2023)).

### 2.7. β-Galactosidase Activity

The β-galactosidase activity was evaluated as previously described [31]. Briefly, growing cells of strain ED5 (*comQ::kan phrC::cat srfA::lacZ*) were supplemented with 50% (*v*/*v*) of the CM of each intestinal isolated *Bacillus* strain (MV4, MV11, MV24, MV30). The growth was followed at 37 °C under aerobic conditions, until the stationary phase of growth. Then, 1 mL of these cultures were harvested and centrifuged at 4000 rpm for 5 min. The pellet was resuspended in 1 mL of Z-buffer (0.06 M Na_2_HPO_4_, 0.04 M NaH_2_PO_4_, 0.01 M KCl, 0.001 M MgSO_4_, and 0.05 M β-mercaptoethanol) and 10 µL of toluene. Tubes were incubated at 30 °C for 20 min and vortexed every 5 min. The enzymatic reaction was started by adding 200 µL of Ortho-nitrophenyl-β-galactoside (ONPG) solution (4 mg/mL ONPG in Z-Buffer without β-mercaptoethanol) and incubated at 30 °C. When the color was clearly yellow (A 420 nm~0.3), the reaction was stopped by adding 500 µL of 1 M Na_2_CO_3,_ and the reaction time was recorded. Tubes were then centrifuged at 10,000 rpm for 5 min and the absorbance at 420 nm was recorded for each sample immediately. The β-galactosidase activity was expressed in Miller Units, as described in [45].

## 3. Results

### 3.1. Preliminary Characterization and Selection of Strains with Probiotic Potentials

Fecal samples of healthy children (Mat. Meth) were collected, heat-treated as previously reported [35], plated on Difco Sporulation (DS) medium, and incubated aerobically at 37 °C for 24–48 h. Putative siblings, colonies of apparently identical morphology originating from the same intestinal sample, were eliminated. All colonies that did not show a free spore or a forming spore still within a sporangium were also eliminated. 

A total of thirty-two aerobic spore formers (MV2-MV33) were selected for further characterization. Cells of all thirty-two strains showed a rod-shaped morphology under the light microscope and were able to form phase-bright ellipsoidal endospores localized in central or sub-terminal positions. Only strain MV19 formed unusual spores that have been described elsewhere [41]. In addition to spores, strains MV19 [41], MV21, MV26, and MV30 formed multiple parasporal granules (possibly inclusion bodies) (Appendix A).

All thirty-two isolates were screened for hemolytic activity on blood agar plates. Isolates appearing greenish were considered able to oxidize the iron of the hemoglobin molecules within red blood cells and indicated as alpha-hemolytic; isolates forming a clear halo around their colonies were considered able to lyse red blood cells and indicated as beta-hemolytic; isolates that did not appear greenish and did not form any halo were considered not hemolytic and indicated as gamma-hemolytic. As shown in Appendix A, ten out of thirty-two isolates were not hemolytic.

The ten not hemolytic strains were analyzed for their resistance to the panel of eight antibiotics indicated as relevant for members of the *Bacillus* genus by the European Food Safety Authority, (EFSA, 2012). As shown in Table 1, four of the ten considered strains (MV4, MV11, MV24, and MV30) were sensitive to all tested antibiotics, four (MV6, MV14, MV20, and MV26) were resistant to a single antibiotic and the remaining two strains were resistant to three (MV15) and four (MV17) antibiotics over the EFSA breakpoint values, respectively. The four strains sensitive to all antibiotics (MV4, MV11, MV24, and MV30) were considered as potential probiotics and used for further characterization.

### 3.2. Species Assignment and Phylogenetic Analysi

The four isolates were used to extract total DNA and the genomic sequences were obtained with a coverage of about 30×. The genomes were approximately 4.0 Mbp long for MV4, MV11, and MV24, and about 5.8 Mbp for MV30 (Table 2). The total number of ORFs was 3860, 3956, and 4380 for MV4, MV11, and MV24, respectively, and higher for MV30 (6003) (Table 2). No plasmid sequences were detected in any of the strains (not shown). All genome sequences were deposited in the GenBank database (Methods).

Based on the analysis of the sequence of the 16S RNA genes, MV4 and MV11 were tentatively considered as belonging to the *B. velezensis* species while MV24 and MV30 were to the *B. subtilis* and *Priestia megaterium* (formerly *Bacillus megaterium*) species, respectively. The Average Nucleotide Identity (ANI) of the four genomes was determined against the genome of a type strain of each of the putative species and confirmed the species assignment suggested by the 16S RNA gene analysis (Table 3). 

The phylogeny of the four strains was further characterized by comparing forty-nine universally conserved genes (Mat.&Meth) with those of several other strains of related species. A phylogenetic tree was generated by Insert Genome into Species Tree app in KBase (Figure 1).

### 3.3. Genome Analyses

MV4, MV11, MV24, and MV30 genomes were screened for the presence of toxin genes previously found in various other *Bacillus* strains [31,41]. As reported in Appendix A, homologs of most of the genes coding for virulence factors were not found in MV4, MV11, MV24, or MV30. Homologs of the *hlyIII* gene, coding for the channel-forming protein Hemolysin III, were found in all four genomes as well as in the non-pathogenic strains *B. clausii* KSM-K16 and *B. subtilis 168* (Appendix A). The two *B. velezensis* strains and MV30 also contained homologs of the *cysC* gene coding for an adenylyl-sulfate-kinase similar to the *Pseudomonas* adenylyl-sulfate-kinase phytotoxin (Appendix A). Only the genome of MV30 contains two homologs of the *cylR2* gene, coding for a DNA binding protein of *Enterococcus* acting as a repressor of the cytolysin operon (Appendix A). 

The functional annotation of the four genomes was performed by using the EggNOG (evolutionary genealogy of genes: Non-supervised Orthologous Groups) database, a public resource classification database able to provide Cluster Orthologous Groups (COGs) of proteins with functional annotations (Appendix A). For all four strains, the majority of the annotated genes were classified into the “Unknown Category” (S category in Appendix A). According to this annotation system, the next most represented classes for all four strains were “amino acid transport and metabolism”, “transcription”, and “carbohydrate transport and metabolism” (respectively, E, K, and G categories in Appendix A).

The genomes of the four isolates were also annotated using the Carbohydrate-Active enZymes (CAZymes) database in comparison with the genome of a reference strain of the respective species. MV4 and MV11 showed a number of genes putatively coding for glycosyl-hydrolase (GH) and a total number of CAZymes bigger than a reference strain of *B. velezensis* (Table 4). MV24 and MV30 showed, respectively, a slightly bigger and smaller number of CAZymes than their reference strains (Table 4). The GH category contains various hydrolases that act on the glycosidic bond and is divided into families. Some families such as GH1, GH4, GH5, GH16, and GH32 contain enzymes involved in the degradation of cellulose, starch, and sucrose while the GH26, GH30, GH43, GH51, and GH53 contain enzymes specialized in the degradation of hemicellulose, xylan, and cellobiose [46]. Figure 2 reports the abundance of the GH families mainly involved in the degradation of plant polysaccharides found in the four MV genomes. With respect to a reference strain of the same species, MV4, MV11, and MV30 contain more genes for the GH13 family and fewer genes for the GH1 family, respectively, containing enzymes specialized in the degradation of starch or monosaccharides (Figure 2).

MV30 also does not contain GH26 and GH43, present in the *P. megaterium* reference strain and coding for enzymes involved in the degradation of mannans or arabinose/xylose, respectively. The GH profile of the *B. subtilis* strain MV24 was very similar to that of the reference strain with the only difference in the number of GH26 (degradation of mannans) higher in MV24 than in *B. subtilis 168* (Figure 2). 

The genome sequences of the four strains were also analyzed to search for homologs of NK (EC 3.4.21.62), using as a query the protein sequence found in *Bacillus subtilis subsp. natto* BEST195 (accession No. BAI84580.1). Homologs were present in MV4 (85.86% identity, MV4_3335), MV11 (86.13%, Contig_6_158), and MV24 (99.21%, MV24_0328), while no homologs of the NK coding gene were present in the MV30. 

The presence of homologs of the GAD, an enzyme that converts glutamate to GABA, was also investigated and a homolog with an identity of 77.9% with the GAD of other *Bacillus* species (WP_049107856.1) was found only in the MV30 genome (MV30_2943). 

### 3.4. Antimicrobial and Antioxidant Activities

The four genomes were analyzed to identify genes coding for antimicrobials and antioxidants. The antiSMASH (Antibiotics and Secondary Metabolite Analysis Shell) database was used to assess the presence of clusters of genes associated with the synthesis of antimicrobials and secondary metabolites. As reported in Appendix A and schematically summarized in Figure 3, various gene clusters were identified in all four strains. In particular, seven, five, and four clusters with at least 80% similarity with known genes were found, respectively, in MV4, MV11, and MV24 while no homologous clusters were found in MV30 (Appendix A). Based on this, both *B. velezensis* strains potentially produce macrolactin H, bacillaene, fengycin, bacillibactin, and bacilysin while only MV4 also codes for difficidin and surfactin while the genome of the *B. subtilis* strain MV24 codes for bacillibactin, bacilysin, subtilosin A, and fengycin (Appendix A). In addition, several gene clusters of all four strains showed low or no similarity with previously identified genes, suggesting that they may code for variants of known molecules or totally new antimicrobials (Appendix A). As an example, clusters coding for surfactin homologs of MV4 and MV11 were compared with the well-characterized surfactin locus of *B. velezensis* FZB42 [47]. The surfactin cluster of MV4 (82% similarity with that of FZB42) and the two surfactin clusters of MV11 (47 and 39% of similarity with that of FZB42) were predicted to produce very different molecules (Figure 4), confirming that the exploitation of these genomes may provide useful information for the identification of novel antimicrobials.

To validate the genomic predictions on the potential production of antimicrobials, the MV strains were tested against a panel of target microorganisms by agar diffusion assay with cell-free supernatants. MV30 did not show any antimicrobial activity against the target bacteria while the other three strains were active against *L. monocytogenes* and *B. cereus*. Only strains of the *B. velezensis* species (MV4 and MV11) were weakly active against some other bacteria and a strain of *Candida albicans* (Table 5). 

All four genomes were screened for the presence of genes coding for antioxidants. Homologs of the enzymes catalase (CAT; EC:1.11.1.6) and superoxide dismutase (SOD; EC:1.15.1.1) were present in all four genomes. In particular, MV4, MV11, and MV24 contain three CAT (MV4_0856, MV4_0864, MV4_1319 in MV4; Contig_4_193; Contig_4_217 Contig_6_12 in MV11; MV24_3061, MV24_0468, MV24_1926 in MV24) and three SOD (MV4_0856, MV4_1319, MV4_0864 in MV4; 1_870, 2_130, 1_878 in MV11; MV24_2302, MV24_1406, MV24_2295) homologs. MV30 genome only contains one homolog of each enzyme (CAT:MV30_1299 and SOD:MV30_1877). 

The four strains were then tested for their antioxidant activity. Vegetative cells (collected after 24 h of aerobic growth at 37 °C) (Methods) were tested for the scavenging activity against hydrogen peroxidase (H_2_O_2_) or free radicals (by α,α-diphenyl-β-picrylhydrazyl (DPPH) method), as previously reported [20]. As shown in Figure 5, the H_2_O_2_-scavenging activity was stronger than the anti-free radicals activity with between 80 and 90% of H_2_O_2_ elimination with 1 × 10^8^ cells and a total H_2_O_2_ elimination with 1 × 10^9^ cells of all four strains. Over 60% of free radical elimination was observed with 1 × 10^9^ cells of MV4, MV11, and MV24 (Figure 5A–C), while about 40% of free radicals reduction was observed with the same number of MV30 cells (Figure 5D).

### 3.5. Matrix Formation

In *B. subtilis*, the model system for spore formers, the biofilm matrix components include secreted proteins, extracellular DNA (cDNA), and an exopolysaccharide (EPS) [48]. The genes coding for the secreted proteins and the EPS are clustered in the *epsA-O* and *tapA-sipW-tasA* operons that are controlled by the master regulators SinI-SinR and Spo0A [48]. Also, in *B. cereus* products of the *pur* operon, coding for enzymes needed for purine biosynthesis is required for biofilm formation [49]. The genomes of MV4, MV11, MV24, and MV30 were analyzed for the presence of all these genes involved in matrix formation. As reported in Appendix A, all four genomes contain homologs of the *epsA-O* and *tapA-sipW-tasA* operons, of their regulators, and of the *pur* operon. 

All four strains were analyzed for their ability to produce biofilm in different media in comparison with a *B. subtilis* laboratory collection strain producing minimal amounts of biofilm (PY79) and a natural strain of the same species known to produce large amounts of biofilm (NCIB3610) [50]. In our experimental conditions (aerobic growth at 37 °C) all four strains produced different amounts of matrix in different media (Figure 6). The two *B. velezensis* strains produced a low amount of biofilm in rich (LB) medium and higher amounts in sporulation-inducing (DSM) or minimal (LB) media (Figure 6). Similarly, to the reference strain of *B. subtilis* NCBI3610, *B. subtilis* MV24 produced the highest amount of matrix in a minimal (S7) medium. However, while the amount of biofilm produced by the two strains was almost identical in LB and DSM media, in minimal medium MV24 produced significantly more biofilm than NCIB3610 (Figure 6). MV30 showed a different pattern of biofilm formation with the highest amount produced in the LB medium and the lowest in the sporulation-inducing medium (Figure 6).

### 3.6. Production of CSF or CSF-like Peptides

Several *Bacillus* strains produce and secrete peptides sensed by human epithelial cells, such as the CSF pentapeptide of *B. subtilis* which is able to induce a cytoprotective heat-shock response [28,30]. The four MV strains were evaluated for their ability to induce a heat-shock response in human epithelial cells. HT-29 colon carcinoma cells were either heat-shocked (30 min at 45 °C) or treated with 20% *v*/*v* of cell-free conditioned medium (CM) from the four MV strains and Hps27 induction evaluated by Western blotting with anti-Hsp27 antibody (Figure 7). 

After normalization of the intensity of the obtained signals with those of the β-actin extracted from the same cells, the levels of Hsp27 after the treatment with the CM of each of the four strains appeared higher than the threshold level obtained with uninduced HT-29 cells (Figure 7). Although the experiment was not quantitative, it suggested that the CM of all four MV strains was able to induce the expression of the heat-shock protein Hsp27 in the HT-29 cells and, therefore, that all four strains secreted CSF or CSF-like peptides.

A quantitative evaluation of the amount of secreted CSF was obtained by using a recombinant strain of *B. subtilis* carrying a null mutation in the gene coding for CSF (*phrC*) and a gene fusion between the promoter of the CSF-controlled gene *rapA* and the coding part of the reporter gene *lacZ* of *Escherichia coli* [30]. The recombinant strain ED5, not producing CSF, does not express the gene fusion but responds to exogenous CSF producing β-galactosidase (Figure 8A) [30]. Cells of ED5 growing in LB medium were supplemented (20% *v*/*v*) with the CM of the four MV and control strains, and tested for β-galactosidase production, as previously reported [30]. The *B. subtilis* isolate MV24 induced β-galactosidase at levels about two-fold higher than the laboratory collection strain of the same species (PY79) (Figure 8B). As expected, the three strains not belonging to the *B. subtilis* species did not induce β-galactosidase, suggesting that they produce CSF-like peptides able to induce a heat-shock response in epithelial cells [30] (Figure 5) but are unable to activate *rapA* transcription in *B. subtilis* (Figure 8B). 

## 4. Discussion

Four aerobic spore former strains isolated from the intestinal samples of healthy children were selected as potential probiotics for not being hemolytic and sensitive to all antibiotics indicated as relevant for spore formers by the EFSA. The four strains were unambiguously assigned to species that are in the Qualified Presumption of Safety (QPS) list of the EFSA (https://www.efsa.europa.eu/en/topics/topic/qualified-presumption-safety-qps (accessed on 5 September 2022)) and are marketed as probiotics for human or animal use [9,12], therefore indicating that they could be considered as safe for human use. This assumption is supported by the in-silico observation that the four MV genomes do not carry putative genes coding for known *Bacillus* toxins. Indeed, the only gene present in all four genomes coding for a putative toxin is the *hlyIII* gene, common to pathogenic and non-pathogenic bacteria. The product of *hlyIII* was, however, not sufficient to cause an alpha- or a beta-hemolytic phenotype in the four MV strains. A gene coding for a putative phytotoxin, *cysC*, is carried by MV4, MV11, and MV30 while only the latter strain contains the *cylR2* gene, a homolog of a transcriptional repressor of a cytolysin operon of *Enterococcus*.

All four strains formed biofilm and showed antioxidant activity in vitro. Both these properties are relevant for a probiotic strain, favoring the colonization of the intestinal environment and protecting the intestinal cells [20]. In addition, MV4, MV11, and MV30 have a gene coding for an NK homolog while MV30 has a gene coding for the enzyme GAD, able to convert glutamate into GABA. NK is a serine protease of the subtilisin family that when orally ingested exhibits fibrinolytic, antithrombotic, and antihypertensive effects [50]. GABA is an inhibitory neurotransmitter that improves various physiological illnesses, including diabetes, hypertension, and depression in humans [51]. Therefore, the potential to produce NK and/or GABA is considered a relevant property for a potential probiotic strain.

The four MV strains considered in this study also produced and secreted CSF (MV24) or CSF-like (MV4, MV11, and MV30) peptides. In addition to inducing anti-inflammatory and cytoprotective responses both in vitro and in vivo [28,29], these molecules have a role in the anti-neurodegenerative activity of *B. subtilis* in the model organism *C. elegans* [31,32,33]. Ingested spores of strains producing such molecules could have beneficial effects for the treatment of inflammatory and age-related neurodegenerative diseases.

Although the production of antimicrobials is not the main mechanism of action of probiotics against infectious diseases, the observed production of an array of secondary metabolites with potential antimicrobial activity by the four strains is certainly an additional relevant point for a probiotic to be used as an alternative to antibiotics. In this context, the identification of genes coding for products with low or no similarity with known antimicrobials is particularly promising. Such genes may encode either variants of previously identified molecules with different targets and mechanisms of action or totally new antimicrobials. The detailed characterization of these genes, which will be a future research challenge, is expected to provide useful information for the identification of novel antimicrobials.

MV4, MV11, MV24, and MV30 strains do not pose safety concerns and have beneficial properties, therefore they can be considered for in vivo experiments aimed at testing their role as an alternative to antibiotics for humans and animals.

## Figures and Tables

**Figure 1 microorganisms-11-01978-f001:**
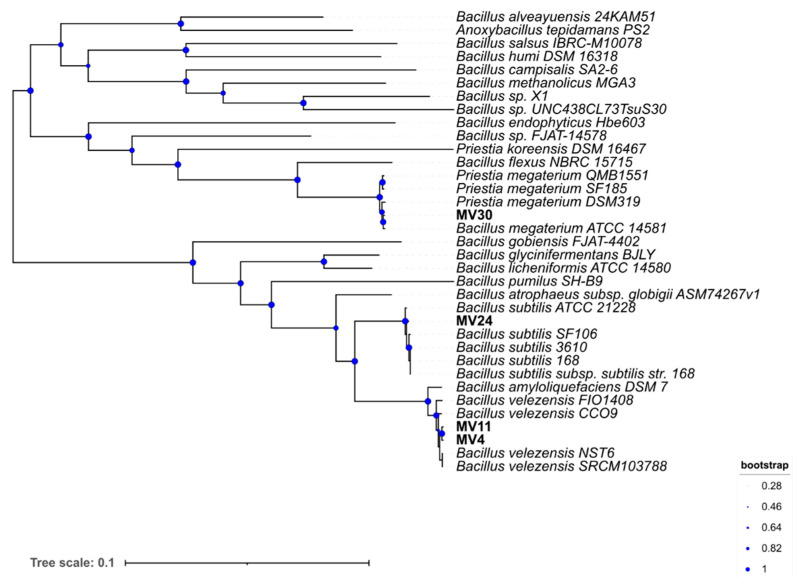
Phylogenetic tree. The genomes were imported into the Kbase system and the phylogenetic tree was constructed using Insert Genome Into SpeciesTree-v2.2.0. Bootstrap confidence values were generated using 1000 permutations.

**Figure 2 microorganisms-11-01978-f002:**
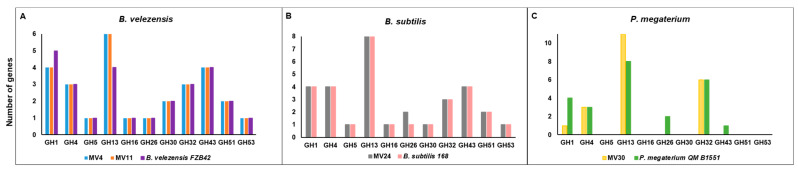
Number of predicted genes encoding cellulose/hemicellulose degradation enzymes found in the genome of MV4, MV11 (**A**), MV24 (**B**), and MV30 (**C**) compared to the indicated reference strains.

**Figure 3 microorganisms-11-01978-f003:**
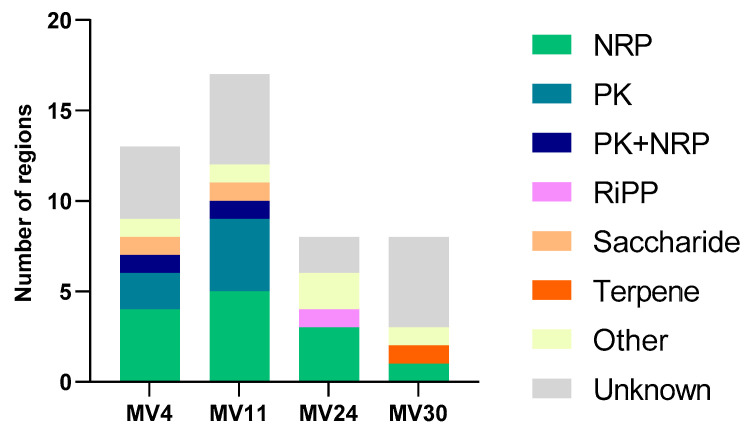
Abundance of clusters of genes coding for antimicrobials and other secondary metabolites in MV4, MV11, MV24, and MV30. The identified clusters were divided into functional classes. NRPS = Non-ribosomal peptide synthetase; PK = polyketide synthase; RiPP = Ribosomally synthesised and post-translationally modified peptide product; Saccharide = possible saccharide; Terpene = possible terpene; Other = Cluster containing a secondary metabolite-related protein that does not fit into any other category; Unknown = unknown BCG clusters.

**Figure 4 microorganisms-11-01978-f004:**
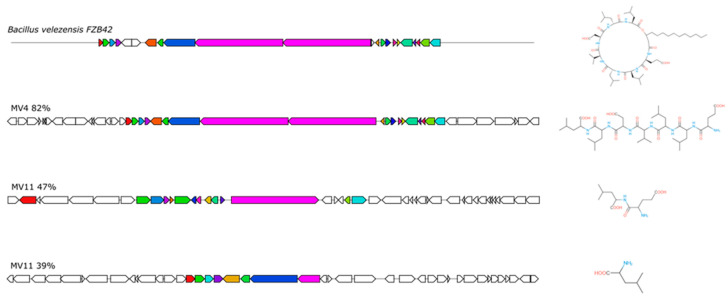
Surfactin loci of MV4, MV11, and the reference strain FZB42. The same colors indicate genes coding for products with similar functions. On the right is reported the predicted structure of each gene cluster.

**Figure 5 microorganisms-11-01978-f005:**
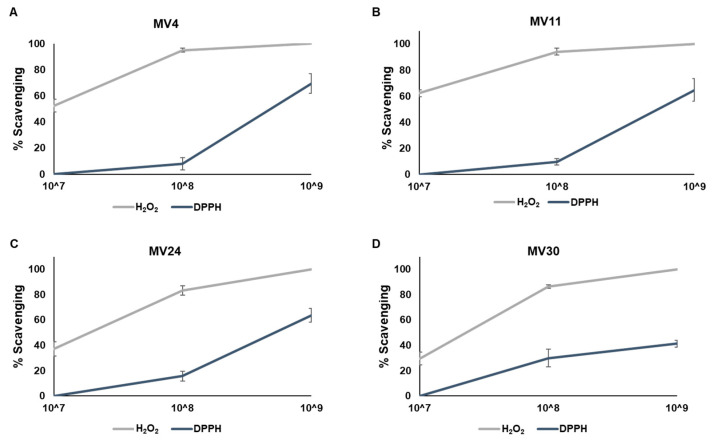
Evaluation of the ability of MV4 (**A**), MV11 (**B**), MV24 (**C**), and MV30 (**D**) cells to reduce hydrogen peroxide (H_2_O_2_, grey line) and the radical 2,2-diphenyl-1-picrylhydrazyl (DPPH, blue line). The assay was performed using suspensions of 10^7^, 10^8^, and 10^9^ cells and was repeated in triplicate.

**Figure 6 microorganisms-11-01978-f006:**
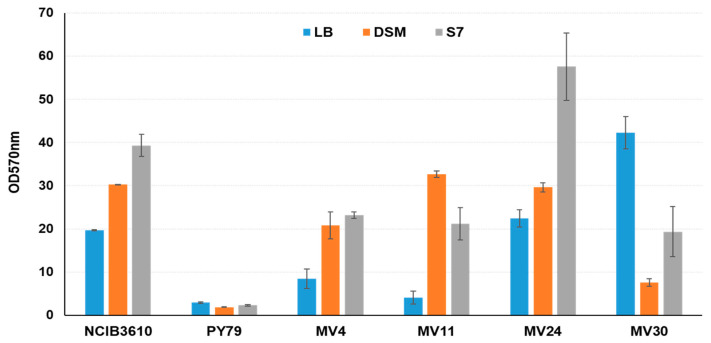
Biofilm production assay performed on the four bacterial strains in three different media: LB (blue bars), DSM (orange bars), and S7 (grey bars). The assay was performed in triplicate. *B. subtilis* strains NCIB3610 and PY79 were used as positive and negative controls, respectively.

**Figure 7 microorganisms-11-01978-f007:**
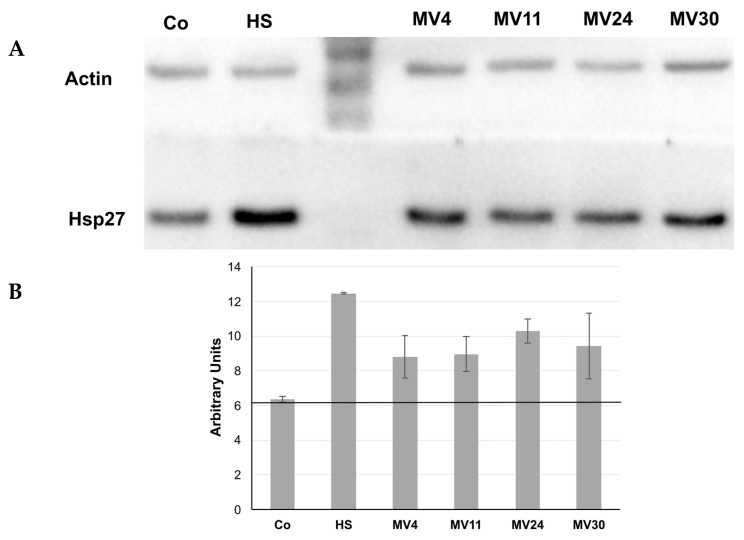
Effect of the conditioned medium (CM) of the four strains on Hsp27 production in HT-29 intestinal cells. Representative Western blot was performed with anti-Hsp27 antibodies (**A**) and a densitometric analysis was obtained by the average of three different experiments (**B**). Untreated (Co) and heat-shocked (HS) cells were used as a negative and positive control, respectively. The band intensity of each sample was evaluated by Image Quant analysis and normalized for the β-Actin intensity of the same sample. The black continuous line indicates the uninduced level of Hsp27 in HT-29 cells.

**Figure 8 microorganisms-11-01978-f008:**
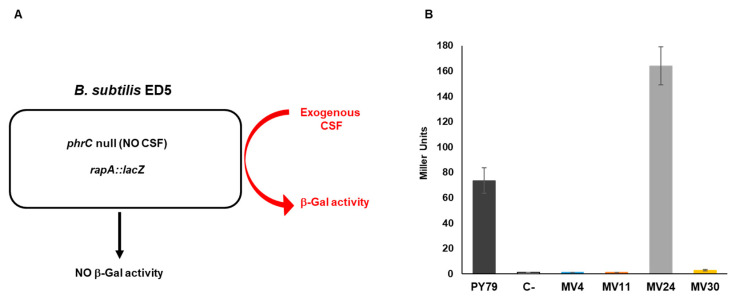
Production of CSF pentapeptide. (**A**) Schematic representation of the mutant strain of *B. subtilis* ED5 unable to produce CSF (*phrC* null) carrying the *rapA::lacZ* gene fusion. The mutant alone does not produce endogenous CSF but responds to exogenous CSF producing β-galactosidase. (**B**) Beta-galactosidase activity was measured with ED5 cells treated with the supernatants of MV4, MV11, MV24, and MV30. Supernatants of PY79 (*B. subtilis* wild-type) and ED5 (isogenic *phrC* deletion mutant) were used as a positive and negative control, respectively. The β-galactosidase is expressed in Miller Units and data are the average of three independent experiments.

**Table 1 microorganisms-11-01978-t001:** Minimum inhibitory concentration (MIC) values in mg/L.

Strain	Van	Gen	Kan	Strep	Ery	Clin	Tet	Chl
MV4	1.500	0.500	0.750	8.000	0.094	0.750	8.000	1.000
MV6	0.500	0.125	2.000	R ^1^	0.125	0.190	1.000	0.250
MV11	0.190	0.750	0.500	1.000	0.047	0.125	2.000	2.000
MV14	3.000	0.125	0.750	8.000	0.250	4.000	R ^1^	5.000
MV15	0.380	0.250	R ^1^	2.000	0.380	R ^1^	0.250	R ^1^
MV17	0.047	0.500	R ^1^	R ^1^	R ^1^	0.750	0.750	R ^1^
MV20	0.190	0.250	0.500	1.000	0.125	R ^1^	1.000	0.750
MV24	1.500	0.500	1.500	3.000	0.380	0.380	0.094	4.000
MV26	0.125	0.047	0.125	1.000	0.064	R ^1^	0.250	2.000
MV30	0.125	0.047	0.190	8.000	0.064	4.000	0.500	1.500
EFSA Breakpoints	4.000	4.000	8.000	8.000	4.000	4.000	8.000	8.000

^1^ R = resistant (over the EFSA breakpoint value).

**Table 2 microorganisms-11-01978-t002:** General features of the MV4, MV11, MV24, and MV30 genomes.

	MV4	MV11	MV24	MV30
Size (bp)	3,987,695	4,035,847	4,116,400	5,853,282
Number of contigs	36	58	112	35
Average Coverage	58	23	41	133
Number of predicted ORFs	3860	3956	4380	6003
Average GC (percentage)	46.35	46.2	43.26	37.46

**Table 3 microorganisms-11-01978-t003:** Average Nucleotide Identity (ANI) (%) based on whole genome alignments.

	*B. velezensis* FZB42	*B. subtilis* 168	*P. megaterium* QM B1551
MV4	98.36	77.3	68.77
MV11	98.36	77.04	68.44
MV24	77.3	98.54	68.97
MV30	68.31	68.58	97.42

**Table 4 microorganisms-11-01978-t004:** Number of genes putatively coding for the six CAZyme categories.

Species and Strain	GH	GT	PL	CE	CBM	AA	Total
*B. velezensis* MV4	47	37	3	11	16	3	117
*B. velezensis* MV11	48	37	3	11	16	3	118
*B. velezensis* FZB42	40	35	3	13	14	5	110
*B. subtilis* MV24	56	39	7	14	22	3	141
*B. subtilis* 168	56	34	7	14	23	4	138
*P. megaterium* MV30	49	42	1	20	23	5	140
*P. megaterium* QM B1551	53	42	1	24	21	5	146

**Table 5 microorganisms-11-01978-t005:** Antimicrobial activity of MV4, MV11, MV24, and MV30 against a panel of target strains.

	MV4	MV11	MV24	MV30
*Listeria monocytogenes* ATCC 7644	+	+	+	-
*Bacillus cereus* ATCC 10987	++	++	+	-
*Enterococcus faecalis* ATCC 29212	-	-	-	-
*Streptococcus faecalis* ATCC 33186	+/-	+/-	-	-
*Escherichia coli* ATCC 25922	+/-	+/-	-	-
*Salmonella enterica* subsp. *enterica serovar* *Typhimurium* ATCC 14028	+/-	+/-	-	-
*Citrobacter rodensis* ATCC 14580	+/-	+/-	-	-
*Shigella sonnei* ATCC 25931	-	-	-	-
*Candida albicans* ATCC 14028	+/-	+/-	-	-
*Lacticaseibacillus rhamnosus* GG ^1^	-	-	-	-

+/- inhibition halo < 1 cm; + inhibition halo > 1 cm e < 2 cm; ++ inhibition halo > 2 cm. ^1^ This representative probiotic strain was used to validate the selectivity of the antimicrobial properties of the isolated bacteria.

## Data Availability

Not applicable.

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
