# Peer review of "Probiotics as an Alternative to Antibiotics: Genomic and Physiological Characterization of Aerobic Spore Formers from the Human Intestine"

_microorganisms, 2023, doi:10.3390/microorganisms11081978_

Round 1

Reviewer 1 Report

The study is relevant and well-planned. The authors showed that all four selected strains contained gene clusters potentially encoding new antimicrobials that exhibited antioxidant activity, formed biofilms, and produced/secreted beneficial peptides. However, there are a few minor remarks.

Please correct the incorrect names of Bacillus velenzensis repeatedly in the text (correct names: Bacillus velezensis), Salmonella enterica typhi (Salmonella enterica subsp. enterica serovar Typhimurium), Lactobacillus rhamnosus (Lacticaseibacillus rhamnosus). Line 421 "gamma" is missing.

Specify abbreviations only at the first mention, and not at the following mentions as well (for example, NattoKinase (NK) and glutamate decarboxylase (GAD) are indicated 2 and 3 times, respectively).

Please reduce the number of self-citations, which is currently (for Ezio Ricca) 15 out of 54, i.e. 28%, which is unacceptable.

Author Response

Reviewer 1

The study is relevant and well-planned. The authors showed that all four selected strains contained gene clusters potentially encoding new antimicrobials that exhibited antioxidant activity, formed biofilms, and produced/secreted beneficial peptides. However, there are a few minor remarks.

REPLY: We thank the reviewer for the positive for the positive consideration of the manuscript. All comments and suggestions have been considered and we feel they helped in improving the quality of the manuscript.

specifically:

1. Please correct the incorrect names of Bacillus velenzensisrepeatedly in the text (correct names: Bacillus velezensis), Salmonella enterica typhi(Salmonella entericasubsp. entericaserovar Typhimurium), Lactobacillus rhamnosus(Lacticaseibacillus rhamnosus). Line 421 "gamma" is missing. REPLY: DONE

2. Specify abbreviations only at the first mention, and not at the following mentions as well (for example, NattoKinase (NK) and glutamate decarboxylase (GAD) are indicated 2 and 3 times, respectively).

REPLY: DONE

3.Please reduce the number of self-citations, which is currently (for Ezio Ricca) 15 out of 54, i.e. 28%, which is unacceptable.

REPLY: we reduced the number of self-citation as suggested (reference number 13, 15 and 27 have been replaced).

Reviewer 2 Report

This manuscript described the characterization of the four aerobic spore forming bacteria MV4, MV11, MV24 and MV30, isolated from intestinal samples of the healthy children, and identification them by 16S rRNA gene sequence, and proved their safety with no hemolytic properties and sensitive to antibiotics tested. By the functional annotation of the four genomes, several probiotic genes such as carbohydrate active enzymes, antimicrobials, antioxidants, glutamic acid decarboxylase, competence and sporulation factor were predicted and experimental data demonstrated their antimicrobial and antioxidant activities, and the cytoprotective effects on TH29 cells by the secreted CSF by MV24. This content of this manuscript was rich. But some issues should be concerned:

1.     Some probiotic properties detected were far away to the topic “ probiotics as alternative to antibiotics” suggesting the content be concise and focused.

2.     The target strains used for determination of the antimicrobial spectrum should be focused on the intestinal pathogenic or other pathogenic bacteria.

Author Response

We thank the reviewer for the suggestions. All comments have been considered and we feel they helped in improving the quality of the manuscript.

Some probiotic properties detected were far away to the topic “probiotics as alternative to antibiotics” suggesting the content be concise and focused.

Reply: This work proposes using the isolated spore-formers bacteria as probiotics and as an alternative to antibiotics. The different properties analyzed, including the absence of hemolytic activity, the sensitivity to antibiotics, the absence of genes encoding for toxins, the belonging of the strains to GRAS species, the antioxidant and cytoprotective activity, and the biofilm production classify these isolated strains as potential probiotic bacteria. Moreover, the antimicrobial activity showed in vitro against twelve selected human pathogenic strains, confirmed by in silico analyses, propose these bacteria as potential alternatives to antibiotics, which will be confirmed by in vivo analyses.

2. The target strains used for determination of the antimicrobial spectrum should be focused on the intestinal pathogenic or other pathogenic bacteria.

REPLY: we thank the reviewer for the useful suggestion. The Table 5 and the text have been modified (line 471). One representative probiotic strain (Lacticaseibacillus rhamnosus GG) was used to validate the selectivity of the antimicrobial properties of the isolated bacteria.

Round 2

Reviewer 2 Report

The authors have corrected my concerns about this manuscript,  I have no comments anymore.